# Photocatalytic Degradation of Some Typical Antibiotics: Recent Advances and Future Outlooks

**DOI:** 10.3390/ijms23158130

**Published:** 2022-07-24

**Authors:** Xue Bai, Wanyu Chen, Bao Wang, Tianxiao Sun, Bin Wu, Yuheng Wang

**Affiliations:** 1Division of Pharmacy and Optometry, Faculty of Biology, Medicine and Health, School of Health Science, University of Manchester, Oxford Road, Manchester M13 9PT, UK; xue.bai@manchester.ac.uk; 2Faculty of Biology, Medicine and Health, School of Health Science, University of Manchester, Oxford Road, Manchester M13 9PT, UK; 16chenj@wycombeabbey.com; 3State Key Laboratory of Biochemical Engineering, Institute of Process Engineering, Chinese Academy of Sciences, Beijing 100190, China; baowang@ipe.ac.cn; 4Helmholtz-Zentrum Berlin für Materialien und Energie GmbH, Albert-Einstein-Straße 15, 12489 Berlin, Germany; tianxiao.sun@helmholtz-berlin.de (T.S.); bin.wu@helmholtz-berlin.de (B.W.)

**Keywords:** antibiotics, photocatalytic degradation, degradation mechanism, photocatalysts

## Abstract

The existence of antibiotics in the environment can trigger a number of issues by fostering the widespread development of antimicrobial resistance. Currently, the most popular techniques for removing antibiotic pollutants from water include physical adsorption, flocculation, and chemical oxidation, however, these processes usually leave a significant quantity of chemical reagents and polymer electrolytes in the water, which can lead to difficulty post-treating unmanageable deposits. Furthermore, though cost-effectiveness, efficiency, reaction conditions, and nontoxicity during the degradation of antibiotics are hurdles to overcome, a variety of photocatalysts can be used to degrade pollutant residuals, allowing for a number of potential solutions to these issues. Thus, the urgent need for effective and rapid processes for photocatalytic degradation leads to an increased interest in finding more sustainable catalysts for antibiotic degradation. In this review, we provide an overview of the removal of pharmaceutical antibiotics through photocatalysis, and detail recent progress using different nanostructure-based photocatalysts. We also review the possible sources of antibiotic pollutants released through the ecological chain and the consequences and damages caused by antibiotics in wastewater on the environment and human health. The fundamental dynamic processes of nanomaterials and the degradation mechanisms of antibiotics are then discussed, and recent studies regarding different photocatalytic materials for the degradation of some typical and commonly used antibiotics are comprehensively summarized. Finally, major challenges and future opportunities for the photocatalytic degradation of commonly used antibiotics are highlighted.

## 1. Introduction

Antibiotics are chemotherapeutic agents that cure bacterial infections [1]. Currently, antibiotics in the environment are attracting increased attention, prompting a widespread search for possible methods of containment [1,2]. This issue results in the generation of antibiotic-resistant genes and antibiotic-resistant bacteria, which expedite the spread of antibiotic resistance, creating a threat to human health and ecological systems [2,3,4]. Thus, in the 21st century, the threat to the integrity of our water resources from antibiotic pollutants is deemed to be one of the most serious environmental problems worldwide, not only because of environmental damage, but due to the potential harm to human health [5].

During the past decade, many strategies have been adopted to address the problem of wastewater antibiotics [6]. Wastewater treatment is usually considered the main method for managing these antibiotics, since wastewater collects discharge from hospitals, industry, and agriculture [7]. However, many more studies have confirmed that conventional treatments are not highly capable of removing these pollutant compounds, which are predominantly water-soluble, and are neither volatile nor biodegradable [8]. Biotic elimination and non-biotic processes, including sorption, hydrolysis, biodegradation by bacteria, and oxidation, as well as reduction, have attracted a great deal of attention [9]. Yang et al. studied the adsorption, desorption, and biodegradation performance of sulfonamide antibiotics in the existence of activated sludge with and without NaN_3_ biocide [10]. The experimental results showed that the antibiotics were eliminated by sorption and biodegradation via the activated sludge. Liu et al. investigated four antibiotics including norfloxacin, ofloxacin, roxithromycin, and azithromycin as target antibiotics, and adopted UV254 photolysis, ozonation, and UV/O_3_ approaches to conduct disposal treatments of nanofiltration, realizing the highest efficiency (>87%) in eliminating antibiotics [11]. Nevertheless, the application of these methods was highly restricted due to the high cost, low stability, and poor recycling ability. Therefore, scientists have been seeking novel methods for degrading antibiotics in wastewater, making the exploration of high-efficiency degradation techniques a popular pursuit for environmental and chemistry researchers [4].

As one of the most promising strategies for degrading antibiotic pollutants, photocatalysis has received much attention due to its low cost, efficiency, and environmental friendliness while degrading antibiotics under sunlight and ambient conditions [12,13]. Most antibiotics are resistant to decomposition owing to their robust molecular structures, thus, the development, design, and fabrication of appropriate photocatalysts with high photocatalytic activities are urgently needed [14]. Though a few catalytic processes have been discussed in the literature so far, there have not been enough examples focusing on the use of appropriate photocatalysts that possess longer wavelength absorption for the photocatalytic degradation of antibiotics, which would help to inform readers about this research field.

The photodegradation of antibiotic pollutants has been reviewed recently [5,15]. However, knowledge of the critical degradation mechanisms and underlying reaction pathways of some typical photodegradation reaction catalysts for antibiotics requires deeper discussion. Furthermore, a comprehensive overview on the possible sources and dangers of the antibiotic pollutants released through the ecological chain, particularly regarding the consequences and damages caused by antibiotic residuals on the environment and human health, is still missing. Additionally, the overall introduction of some commonly-used photocatalytic nanomaterials and their application in the degradation of some typical antibiotics is essential to confirm their practical superiority and effectiveness as photodegradation catalysts.

This review firstly summarizes the effects of antibiotics on living organisms and the environment as well as the basic mechanism of the photocatalytic degradation of antibiotics. Then, commonly used photocatalytic materials for antibiotic degradation are reviewed. Finally, the recent advances in the use of various photocatalytic materials for the degradation of antibiotics are discussed.

## 2. Consequences of Antibiotics in Wastewater on the Environment and Human Health

Pharmaceuticals can largely improve humans’ health and quality of life when used to treat contagious diseases, however, the misuse of drugs, especially antibiotics, has severe damage on the environment and human health [6,16]. Some results reported remarkable changes in sex ratio and fecundity of daphnia manga when exposed to antibiotics such as sulfamethoxazole and trimethoprim [17]. Meanwhile, a decrease in desire and sexual motivation was observed in experiments on male rats given cimetidine [18].

In some countries, antibiotics are not only employed for animal treatment but also to accelerate animal growth and increase production. Thus, antibiotics might be released from animal waste due to incomplete digestion, and that waste may then be used as fertilizer in agriculture or dumped into wastewater, generating a possible pathway to human harm from food or drink exposure, as shown in Figure 1 [19]. A recent study reported that chlortetracycline antibiotic was, to some extent, uptaken by onions, cabbage, and corn [20]. However, those vegetables did not uptake tyrosine antibiotic, probably due to its large molecular size. Thus, continuous release of these antibiotic pollutants into water environments and organisms has a severely negative impact on the environment by causing genetic exchange and activating drug-resistant bacteria. In particular, most antibiotic pollutants, even under low concentrations, may result in a severe risk to the ecosystem and human health [21,22].

On one hand, in terms of micro-organisms, the release of antibiotics into the environment could lead to chromosomal mutations of native bacteria, triggering the development of antibiotic-resistant bacterial strains, which may cause environmental threats such as toxicological effects on non-intended pathogens, alteration of structures, and dissemination of algal communities [23,24,25]. On the other hand, consumption of water or agricultural and sideline products containing antibiotic pollutants may induce symptoms in humans including, but not limited to, vomiting, tremors, nausea, headache, diarrhea, and nervousness [26]. Furthermore, problems such as restraining spinach growth, physiological teratogenesis, and human gene toxicity have also been reported due to the presence of antibiotic pollutants in water or food [3,27].

## 3. Principle and Fundamental Mechanism of Photocatalytic Degradation of Antibiotics

The steps involved in the photocatalytic degradation of antibiotics are demonstrated in Figure 2. The predominant mechanisms for antibiotic photocatalytic degradation can be summarized as three main steps: photon absorption, excitation, and reaction [5,7]. In detail, once a photocatalyst absorbs photons with an energy higher than its band gap, the electrons in the valance band (VB) can be excited and jump up into the conduction band (CB), where a hole (h^+^_VB_) is produced (Equation (1)) [6,26,28,29]. Subsequently, the photogenerated electrons and holes are efficiently separated and migrate to the surface of the photocatalyst, triggering secondary reactions with the adsorbed materials. Typically, photogenerated holes can also attack those antibiotics directly (Equation (2)), theoretically leading to significant degradation of those toxic antibiotics. In addition, two types of systematic theories about degradation pathways were proposed and recognized by researchers in this field [26]. One is a reductive pathway that happens if the CB potential of the semiconductor is negative compared to that of the O_2_/•O_2_^−^ redox potential (−0.13 eV vs. reference hydrogen electrode (RHE)), wherein the photoexcited electrons can react with electron acceptors such as O_2_ deposits on the catalyst surface or dissolved in water, thereby reducing it to form superoxide radical anion •O_2_^−^ (Equation (3)) [6,26]. In contrast, another pathway referring to the oxidative pathway was initiated when the holes migrated to the photocatalyst surface, accompanied by hydroxyl radical (•OH) generation upon the oxidation of H_2_O/OH^−^ depending on the alkalinity or acidity of the media (Equation (4)) [6,26]. After being excited, hydrogen ions could recombine with the electrons and generate heat energy (Equation (5)), which would decrease the efficiency of the photodegradation. It is noted that the standard redox potential of photocatalysts should be higher than that of •OH/OH^−^ (+1.99 eV vs. RHE) in this case [6,26]. Then, both of these reactive radicals (•OH and •O_2_^−^) are highly active oxidizing agents in the photocatalytic process [30]. They can effectively mineralize any antibiotics and their intermediates to form water and carbon dioxide under prolonged exposure to high-energy UV irradiation, and eventually decompose into CO_2_ and H_2_O (Equation (6)) [28,29,31,32]. Many studies demonstrated that both pathways (reductive and oxidative) should synergistically occur to largely prevent the accumulation of electrons in the CB and significantly increase the possibility of the recombination of electrons and positive holes compared to the pathway of direct interaction between photogenerated holes and antibiotics [33].
Photocatalyst + h*υ* → photocatalyst + h^+^ + e^−^(1)
h^+^ + antibiotics → H_2_O + CO_2_ + degradation products(2)
O_2_ + e^−^ → •O_2_^−^(3)
H_2_O/OH^−^ + h^+^ → •OH + H^+^(4)
H^+^ + e^−^ → energy(5)
Antibiotics + •OH or •O_2_^−^ → CO_2_ + H_2_O + degradation products(6)

Considering the prediction of application and efficiency of a type of photocatalytic material, optical bandgap (E_g_) is a very important factor for evaluating photoabsorption ability and photocatalytic efficiency. Mehrorang et al. put forward a method and criterion for bandgap measurement, and divided the concept of the bandgap into the two categories of photonic and electrochemical bandgap facing polyfluorene co-polymers as photocatalysts [34,35]. In addition, They concluded that, the prevention of charge recombination would accordingly lead to a higher lifetime of the active holes, thereby upgrading their antibiotic degradation activity. This proved to be a great strategy for enhancing the activity of photocatalysts under visible light, relating to the interfacial charge transfer from a separate energy surface to a molecular continuous surface from solids [34,35].

All in all, the mechanism of photocatalysis for the degradation of antibiotics can be divided into five main steps: (1) transfer of antibiotics in the fluid phase to the surface; (2) adsorption of the antibiotics; (3) reaction in the adsorbed phase; (4) desorption of the products; and (5) removal of products from the interface region [36,37]. However, photocatalytic degradation suffers the problem of electron-hole recombination in the photocatalyst when the electrons that had been excited to CB rapidly recombine with the separated holes in the VB before creating free radicals [37]. Although this depends on many flexible options such as tuned experimental conditions, the adoption of specific photocatalysts with a low CB–VB bandgap energy and photocatalyst modifications are considered as solutions for these challenges [38,39].

## 4. Common Photocatalytic Materials for Antibiotic Degradation

### 4.1. Semiconducting Metal Oxides-Based Photocatalysts

Metal oxide semiconductors have been utilized as pristine photocatalysts or as hybrids, or have been coupled/doped with other materials to facilitate the degradation of organic pollutants such as pesticides, dyes, and polycyclic aromatic hydrocarbons [40]. More importantly, the application of metal oxide-based photocatalysts for antibiotic degradation has recently drawn more interest and attention from researchers due to their good light absorption under UV, visible light, or both, combined with their biocompatibility, safety, and stability when exposed to different conditions [3,41,42]. Generally, metal oxides encounter some challenges regarding ineffectiveness or non-absorbance of photocatalytic activity because of their wide band gap (Figure 3) and faster electron–hole pair recombination [43]. For example, TiO_2_ is the most popular metal oxide for photocatalysis because of its good optical and electronic properties, chemical stability and reusability, non-toxicity, and low cost [44]. Additionally, ZnO is another semiconducting material that has a better quantum efficiency and higher photocatalytic efficiency compared to TiO_2_, particularly if used for photocatalytic antibiotic degradation at a neutral pH, however, the high recombination rate of the photogenerated electron–hole pairs limits the utilization of ZnO without any functionalization [45]. Several studies demonstrated that doping with metals like Ag and Fe, or non-metals like N and C, into ZnO enhanced the activity of photocatalytic antibiotic degradation [46,47]. WO_3_ is another promising metal oxide that has received remarkable attention due to its abundance, cost-effectiveness, and non-toxicity [48,49]. Furthermore, W_18_O_49_ was also considered a superior photocatalyst with a higher photocatalytic degradation efficiency compared to WO_3_ [3,50]. Nevertheless, it is prone to oxidization to WO_3_ in spite of its superior photocatalytic performance. Thus, the construction of a hybrid of W_18_O_49_ and other metal oxides can overcome this oxidization barrier [3,51]. There are many other metal oxides that play important roles in photocatalytic materials for antibiotic degradation, and we may introduce more in the next section.

### 4.2. Bismuth-Based Photocatalysts

Bismuth, possessing an atomic electron configuration of 6s^2^6p^3^, is a metallic element from the fifth group of the sixth period in the periodic table, and is usually present in the form of Bi^3+^ [26,52]. A class of novel processes along with bismuth-based catalysts have been developed for antibiotic degradation, as shown in Figure 4 [26]. Bi oxides display a narrower bandgap due to the overlap of O 2p and Bi 6s orbitals in the valence band and lone-pair distortion of the Bi 6s orbital, resulting in the mobility of photoexcited charges, enhancing the visible light response performance [53]. Interestingly, the Bi^5+^ valence state from the oxidation of Bi^3+^ has good absorption of visible light once the 6s orbital is empty [54]. Basically, Bi-based photocatalysts mostly have a bandgap of less than 3 eV. There are some typical Bi-based photocatalysts attracting more attention recently, such as Bi_2_O_3_ and BiVO_4_ [55]. Bi_2_O_3_ is one of the most common photocatalysts, showing excellent photocatalytic performance on water-splitting and water treatment from organic wastes [56]. Bi_2_O_3_ has a bandgap ranging from 2.1 to 2.8 eV, making its utilization for visible light absorption more efficient. Bi_2_O_3_ has five different configurations: α, β, γ, δ, and ω-Bi_2_O_3_ [26]. Additionally, BiVO_4_, with superior physicochemical properties like ferro-elasticity and ionic conductivity, has a theoretical bandgap of 2.047 eV, which maximizes its visible light utilization [57]. BiVO_4_ was widely used in photocatalytic reactions for organic waste treatment and water splitting in past years [58]. Although it has been confirmed that bismuth-based photocatalysts have good photocatalytic performance and use visible light efficiently, it should be noted that some parameters, such as stability and solubility, need to be emphasized [59].

### 4.3. Silver-Based Photocatalysts

The application of photocatalytic degradation of silver-based photocatalysts such as AgX (X = Cl, Br, I), Ag_2_O, Ag_3_PO_4_, and Ag_2_CO_3_ have been reported by various researchers [60,61,62]. In the case of pristine Ag_2_CO_3_, the challenge is that pristine Ag_2_CO_3_ is unstable and photocorrosive due to its possible transformation from Ag^+^ to metallic Ag on account of accepted photoelectrons during the photocatalytic processes [63]. Moreover, pristine Ag_2_O also exhibits poor stability and rapid electron-hole recombination [64]. Their superior photocatalytic performance on antibiotic degradation depends not only on the reduced electron-hole recombination but may also be from broad and strong absorption ranges in the visible region due to the localized surface plasmon resonance effects induced by Ag nanoparticles [3,7,65].

### 4.4. Metal-Organic Frameworks (MOFs)-Based Photocatalysts

Metal-organic frameworks (MOFs) are a new class of coordination polymers with periodic network structures formed by the self-assembly between metal ions/metal clusters and organic ligands [66]. By modifying linkers employing functional groups, highly porous structures with remarkable surface areas could be obtained with tuned surface structures [67]. MOFs were first discovered in the mid-1990′s by Omar Yaghi, and the invention of novel MOFs promised long-lived influence in the areas of chemistry, physics, biology, and the material sciences, particularly used extensively in photocatalysis due to their high surface area, adjustable porosity and pore volume [67,68,69]. Thus, MOFs promise to be highly-effective materials for the photocatalytic degradation of antibiotics in a solution [66,70]. Although various MOF-based materials have been utilized to remove antibiotics, the development of more efficient degradation agents remains a key problem facing more active MOF-based photocatalytic degradation materials with more active sites and large surface areas with group functionalization [66,67]. According to the previous report, the organic linker serves as the VB, while the metallic cluster acts as CB. Under exposure to light, MOFs behave like semiconductors, and can thus be deemed as a potential photocatalyst for highly effective degradation of antibiotics due to their superior high thermal and mechanical stability and their excellent structural characteristics [66].

### 4.5. Graphitic Carbon Nitrides-Based Photocatalysts

Graphitic carbon nitride (g-C_3_N_4_), a new class of polymeric semiconducting material, is another kind of promising material for photo-driven catalytic applications [26,71]. On one hand, the past report implied that the g-C_3_N_4_ has a bandgap of around 2.7 eV, and the CB–VB can also meet the requirement of overall water splitting, as demonstrated in Figure 5, which made it more popular for photocatalytic water splitting in the past decades [72].

On the other hand, g-C_3_N_4_ can also be used for the photocatalytic degradation of antibiotic pollutants under visible light. However, pure g-C_3_N_4_ exhibited a low degradation rate due to negative position, resulting in a weak oxidation ability [26,73,74]. Therefore, surface modification is necessary to overcome these limitations [26]. Researchers found that doping with noble metal ions optimized the photocatalytic performance according to several reports in the literature as a result of the higher separation of the photoproduced electrons and holes due to the excellent capacity of electron capture by the noble metallic ions [26,75]. A great deal of research is ongoing regarding g-C_3_N_4_ modification for fabricating and designing nanomaterials with different properties in order to obtain the best possible photocatalytic performance for the removal of antibiotics [76].

## 5. Recent Advances in Photocatalytic Degradation of Antibiotics

### 5.1. Photocatalytic Degradation of Ciprofloxacin

Ciprofloxacin is a second-generation fluoroquinolone antibiotic used to kill bacteria to prevent severe infection [77]. The chemical structure of ciprofloxacin is exhibited in Figure 6a [1]. Notably, ciprofloxacin dominates 73% of the total consumption, with a daily dose between 0.39 and 1.8 per 1000 patients, and has a broad antimicrobial spectrum that has impact on the DNA gyrase and topoisomerase IV of various Gram-positive and Gram-negative bacteria, thus preventing cell replication [78,79]. It has been considered a good therapy for treating digestive infections, complicated urinary tract infections, sexually transmitted diseases, pulmonary diseases, and skin infections. However, the presence of ciprofloxacin limits photosynthetic pathways and even leads to morphological deformities in higher plants. It also leads to severe damage to human health [78]. In the past, various studies reported the use of modified photocatalysts to meet the demand for higher photocatalytic efficiency in the degradation of ciprofloxacin [80].

Yu et al. [80] prepared Zn-doped Cu_2_O particles by a solvothermal method to achieve photocatalytic degradation of ciprofloxacin. The photocatalytic results demonstrated that Zn-doped Cu_2_O has better photocatalytic performance and reusability compared to the undoped Cu_2_O. 94.6% of ciprofloxacin was degraded in presence of Zn-doped Cu_2_O, even after 5 cycles, the degradation percentage still remains 91% due to the significantly enhanced absorption intensity in the visible light range, and the increased band gap than that of the undoped Cu_2_O (Figure 7a,b). In addition, a novel Z-scheme CeO_2_–Ag/AgBr photocatalyst was fabricated by Malakootian et al. [77] using in situ interspersals of AgBr on CeO_2_ for subsequent photoreduction process. The results also exhibited largely enhanced photocatalytic activity for the photodegradation of ciprofloxacin under visible light irradiation due to the faster interfacial charge transfer process and the largely enhanced separation of the photogenerated electron-hole pairs. Furthermore, Pattnaik et al. [81] adopted exfoliated graphitic carbon nitride into photocatalytic degradation of ciprofloxacin under solar irradiation and catalytic data have shown that photocatalytic activities of g-C_3_N_4_ have enhanced after its exfoliation because of its efficient charge separation, low recombination of photogenerated charge carriers and high surface area. They found that 1 g/L exfoliated nano g-C_3_N_4_ can degrade up to 78% of a 20 ppm solution exposed to solar light for lasted 1 h (Figure 7c,d).

### 5.2. Photocatalytic Degradation of Tetracycline

Tetracyclines are a series of broad-spectrum antibiotics that were first adopted in 1940, and their structures are shown in Figure 6b [1,82]. All tetracyclines have anti-inflammatory and immunosuppressive effects, and were previously used to treat rheumatism [83]. Due to the additional effect of tetracyclines against lipases and collagenases, these antibiotics were also initially used for the intrapleural treatment of malignant effusions [84,85]. Although tetracycline plays a significant role in medicine, the existence of tetracycline in aquatic media is of great concern because of its ecological impact, including carcinogenicity and toxicity to the environment [84,85]. A number of studies related to the removal or degradation of tetracycline through the use of different photocatalytic materials have been reported in past years [86]. For example, a novel TiO_2_/g-C_3_N_4_ core-shell quantum heterojunction prepared by a feasible strategy of polymerizing the quantum trick graphitic carbon nitride (g-C_3_N_4_) onto the surface of anatase titanium dioxide nanosheets was put forward by Wang et al. to be employed as a tetracycline degradation photocatalyst, and this catalyst exhibited the highest tetracycline degradation rate: 2.2 mg/min, which is 36% higher than that of the TiO_2_/g-C_3_N_4_ mixture, 2 times higher than that of TiO_2_, and 2.3 times higher than that of bulk g-C_3_N_4_ (Figure 8a,b) [87]. Moreover, Wang et al. synthesized a novel C–N–S tri-doped TiO_2_ using a facile and cost-effective sol–gel method with titanium butoxide as titanium precursor and thiourea as the dopant source, which can be used for photocatalytic degradation of tetracycline under visible light [86]. The catalytic results exhibited the highest photocatalytic degradation efficiency of tetracycline under visible light irradiation which is associated with the synergistic effects of tetracycline adsorption due to its high surface area, narrow band gap causing C–N–S tri-doping, presence of carbonaceous species functioning as a photosensitizer, and well-organized anatase phase. Additionally, Chen et al. synthesized a novel heterostructured photocatalyst AgI/BiVO_4_ by an in situ precipitation procedure, and the results exhibited excellent photoactivity for tetracycline decomposition under visible light irradiation, the tetracycline molecules were apparently eliminated (94.91%) within 60 min, and degradation efficiency was remarkably superior to those of bare BiVO_4_ (62.68%) and AgI (75.43%) under same experimental conditions (Figure 8c,d) [88].

### 5.3. Photocatalytic Degradation of Norfloxacin

Norfloxacin is another antibiotic within the fluro-quinolones group, and is widely used for curing urinary tract infections [89]. Figure 6c indicates the chemical structure of norfloxacin [1]. Currently, the presence of norfloxacin in wastewater (especially in hospitals) contains high concentrations and is deemed to be one of the potential pollutants in the aquatic environment. In the past few years, fluoroquinolone antibiotics have triggered tremendous concern due to their widespread use and environmental toxicity [89,90]. There are several reports on the degradation of norfloxacin by photocatalysis using different materials [91]. Sayed et al. [92] prepared a novel immobilized TiO_2_/Ti film with exposed {001} facets via a facile one-pot hydrothermal route to use in the degradation of norfloxacin from aqueous media, and the experimental results demonstrated excellent photocatalytic performance toward the degradation of norfloxacin in various water matrices, with the observation that •OH is mainly involved in the photocatalytic degradation of norfloxacin by {001} faceted TiO_2_/Ti film (Figure 9a). Additionally, Tang et al. [93] realized excellent visible-light-driven photocatalytic performance for the degradation of norfloxacin by an as-prepared novel Z-scheme Ag/FeTiO_3_/Ag/BiFeO_3_ using a sol–gel method followed by a photo-reduction process. The results showed the photocatalytic degradation extent reaches 96.5% within 150 min when using Ag/FeTiO_3_/Ag/BiFeO_3_ at 2.0 wt.% Ag (FeTiO_3_: BiFeO_3_ = 1.0:0.5) which can be reused with excellent photocatalytic stability (Figure 9b–d). Moreover, Lv et al. [94] synthesized copper-doped bismuth oxybromide (Cu-doped BiOBr) using a solvothermal method, and assessed their ability to degrade norfloxacin under visible light. The as-prepared Cu-doped BiOBr showed high activity with a photocatalytic degradation constant of 0.64 ×10^−2^ min^−1^ in the photocatalytic degradation of norfloxacin under visible-light irradiation due to its enhanced light-harvesting properties, enhanced charge separation, and interfacial charge transfer, as well as a retention of 95% of its initial activity, even after 5 constant catalytic cycles. Similarly, Bi_2_WO_6_, another bismuth-based catalyst, was put forward by Tang et al. [95] and applied to the photodegradation of norfloxacin in a nonionic surfactant Triton-X100 (TX100)/Bi_2_WO_6_ dispersion under visible light irradiation. The results found that the degradation of barely insoluble norfloxacin could be strongly enhanced with the addition of TX100. TX100 was adsorbed strongly on the Bi_2_WO_6_ surface and promoted norfloxacin photodegradation at the critical micelle concentration (CMC = 0.25 mM).

### 5.4. Photocatalytic Degradation of Amoxicillin

Amoxicillin is a penicillin-type antibiotic medicine extensively used for the treatment of various bacterial infections such as dental infections, chest infections, and other infections (ear, throat, and sinus). Figure 6d exhibits the chemical structure of amoxicillin [1]. However, amoxicillin in water or in the ecological environment is considered an emerging pollutant because it can cause several health effects to aquatic life in the presence of solved molecules in water [16]. There are some reports regarding the use of different photocatalytic materials to degrade amoxicillin [96]. For instance, the photocatalytic degradation of amoxicillin by as-prepared titanium dioxide nanoparticles loaded on graphene oxide (GO/TiO_2_) by the chemical hydrothermal method was evaluated under UV light by Balarak et al., and the experimental data exhibited that key indexes such as initial pH, GO/TiO_2_ dosage, UV intensity, and initial amoxicillin concentration all had a significant impact on amoxicillin degradation (Figure 10a) [97]. The efficiency of amoxicillin degradation collection was measured to be more than 99% at specific conditions, including a pH of 6, a GO/TiO_2_ dosage of 0.4 g/L, an amoxicillin concentration of 50 mg/L, and an intensity of 36 W. Additionally, Mirzaei et al. [98] synthesized a new fluorinated graphite carbon nitride photocatalyst with magnetic properties by a gentle hydrothermal method that can be used for the degradation of amoxicillin in water. Compared to the bulk g-C3N4, magnetic fluorinated Fe_3_O_4_/g-C_3_N_4_ with a high specific surface area (243 m^2^g^−1^) resulted in improved photocatalytic activity regarding amoxicillin degradation and the mineralization of the solution. Furthermore, Huang et al. [99] also prepared novel carbon-rich g-C_3_N_4_ nanosheets with large surface areas by a facile thermal polymerization method, which displayed superior photocatalytic activity for amoxicillin degradation under solar light (Figure 10b–d). Meanwhile, the catalyst showed high stability and amoxicillin degradation ability under various media conditions, indicating its high applicability for amoxicillin treatment.

Finally, we list a table to compare the different photocatalysts discussed above, and note that the information in Table 1 shows most of the catalysts can easily and effectively remove the antibiotic contaminations within 2 h with relatively high degradation efficiency, which fully proves the superiority of rapid and effective antibiotic removal by photodegradation.

## 6. Conclusions and Future Perspective

In this review, the photocatalytic degradation of antibiotics was summarized. Firstly, the mechanism of photocatalytic degradation of antibiotics depending on the formation of free radicals and active oxygen species, and the consequences of antibiotics in wastewater on the environment and human health were reviewed. Some widely used antibiotics were then analyzed, and a number of commonly used photocatalysts were introduced. Heteroatom doping is generally used as a strategy to enhance the photocatalytic performance of a photocatalyst, particularly metal atoms as dopants. However, it should be noted that metal dopants could serve as recombination centers at higher concentrations, which can reduce the efficiency of a photocatalyst. Consequently, future research should also focus on other options, including doping with non-metals such as nitrogen, boron, sulfur, and phosphorus. Meanwhile, the formation of a heterojunction with other semiconductors can also play a significant role in the modification of photocatalysts on the degradation of antibiotics due to other semiconductors possibly serving as photosensitizers while simultaneously inhibiting electron-hole recombination. Thus, these methods can achieve visible light-driven photocatalysts with enhanced photocatalytic activity by narrowing the band gap of the photocatalyst or by increasing the activity of charge separation.

Physicochemical properties such as morphology and surface areas are also very critical factors in the performance of catalysts during photodegradation studies. As mentioned above, further studying photocatalysts with different morphologies and surface areas can effectively enhance the performance of catalysts. Furthermore, the degradation pathway also provides a clear introduction to the fate and transformation of antibiotics during the photocatalytic degradation process. Thus, exploring the photocatalytic degradation mechanism at the atomic level is also necessary for accelerating the efficiency of antibiotic degradation.

The utilization of solar radiation and visible light sources to activate photocatalysts during the photodegradation of antibiotics such as ciprofloxacin, tetracycline, norfloxacin, and amoxicillin is still limited. Therefore, the exploration and development of photodegradation induced by UV light sources are still in urgent demand.

Firstly, in the long term, although the removal rate of antibiotics is still being optimized, the removal rate of the chemical oxygen demand is still relatively high during the degradation of antibiotics by photocatalysts, and thus it confirms that the mineralization degree of antibiotics needs to be optimal. There are many intermediates during the process of photodegradation, therefore a deep study into intermediates is also critical for improving the performance of catalysts. Secondly, most experiments involve regular and constant stirring to prevent the agglomeration of materials in media during the degradation of antibiotics by photocatalysts, which requires additional energy consumption. Thirdly, the problems of antibiotics are not only induced by water quality but also by the accumulation of antibiotics in water, which lead to the generation of microbial resistance genes. There is currently still a lack of research on photocatalysts’ limited resistance genes. Finally, The recycling ability of a photocatalyst is a significant index for evaluating its cost-effectiveness and feasibility for practical application in the degradation of antibiotics. To minimize any possible waste, the design of photocatalysts with quasi-same photoactivity during each cycle is preferred. Also, it is important to design photocatalysts that are easier to separate and recycle in order not to avoid losing any worthy materials during the photocatalytic reaction. Thus, the separation of photocatalysts from the aqueous phase is crucial from an economic standpoint. It is noted that the operating cost of a photocatalytic reaction mainly originates from its being a single-use photocatalyst, unable to be recycled. Regarding the repetitive usage of photocatalysts, deep research on how to sustainably use recyclable photocatalysts for antibiotic degradation is still urgently needed.

## Figures and Tables

**Figure 1 ijms-23-08130-f001:**
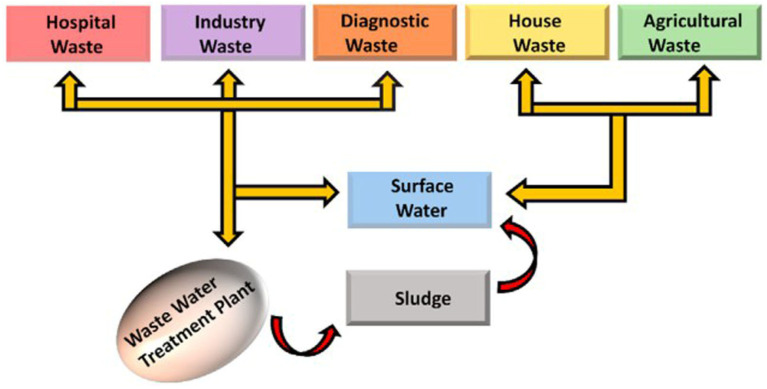
Sources of different emerging antibiotics pollutants in daily life. (Reproduced with permission from [19], copyright 2021, Elsevier).

**Figure 2 ijms-23-08130-f002:**
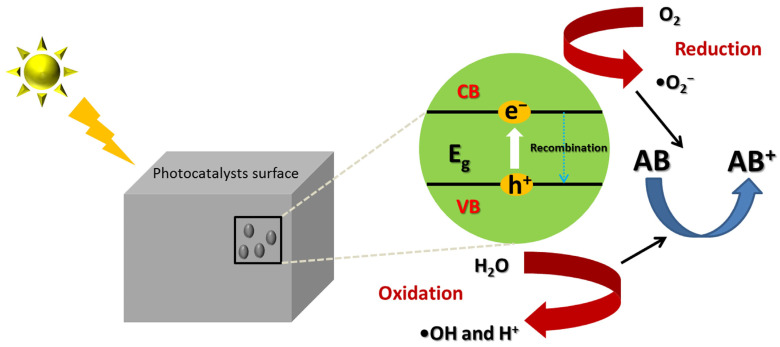
General photocatalytic mechanism on the degradation of antibiotics by the formation of photo-induced charge carriers (e^−^/h^+^) on the photocatalysts’ surface.

**Figure 3 ijms-23-08130-f003:**
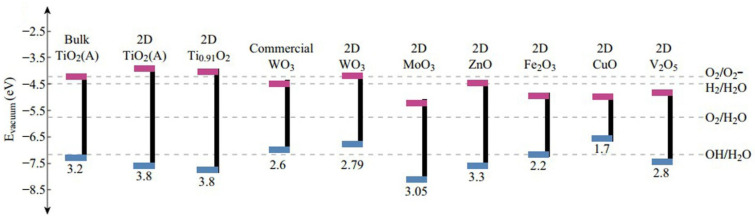
Band energy gaps of selected semiconductors. (Reproduced with permission from [43], copyright 2017, Springer Nature).

**Figure 4 ijms-23-08130-f004:**
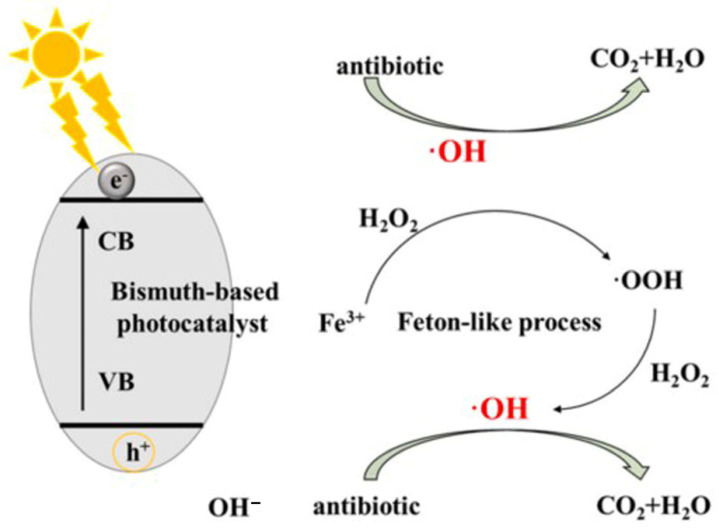
The mechanism of coupled processes with bismuth-based compounds for antibiotic degradation. (Reproduced with permission from [26], copyright 2021, Elsevier).

**Figure 5 ijms-23-08130-f005:**
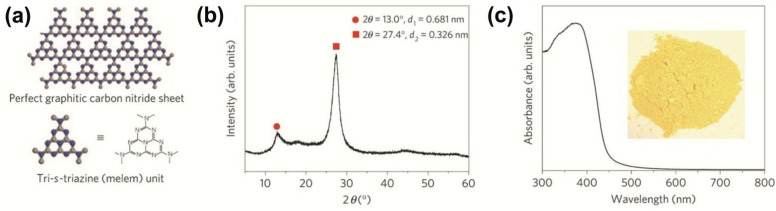
Crystal structure and optical properties of graphitic carbon nitride: (**a**) Schematic diagram of a perfect graphitic carbon nitride sheet constructed from melem units, (**b**) Experimental XRD pattern of the polymeric carbon nitride, revealing a graphitic structure with an interplanar stacking distance of aromatic units of 0.326 nm and (**c**) Ultraviolet-visible diffuse reflectance spectrum of the polymeric carbon nitride. Inset: Photograph of the photocatalyst. (Reproduced with permission from [72], copyright 2009, Springer Nature).

**Figure 6 ijms-23-08130-f006:**
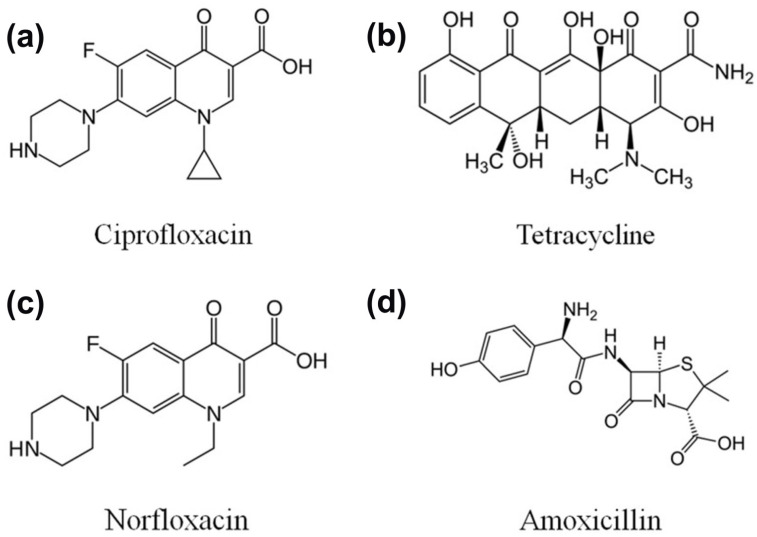
The illustration of commonly investigated antibiotics in photocatalytic processes. (**a**) Ciprofloxacin. (**b**) Tetracycline. (**c**) Norfloxacin. (**d**) Amoxicillin. (Reproduced with permission from [1], copyright 2021, Elsevier).

**Figure 7 ijms-23-08130-f007:**
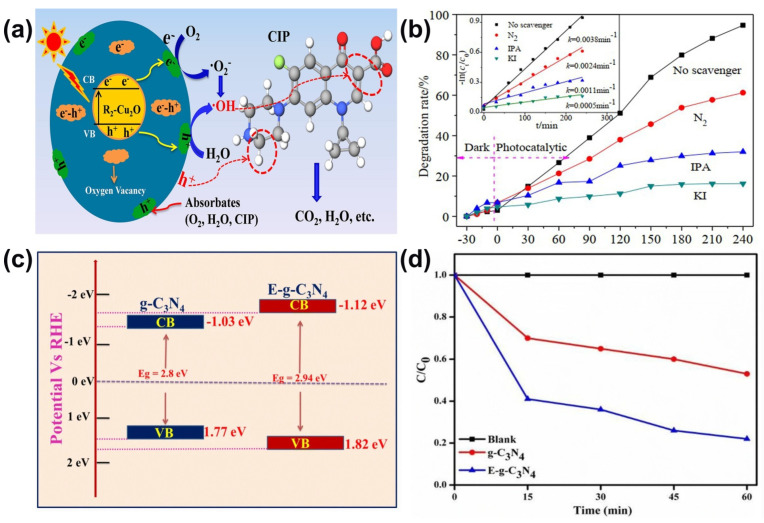
(**a**) The schematic diagram of the photocatalytic mechanism for the Zn-doped Cu_2_O (**b**) Photocatalytic activity of R_2_-Cu_2_O with different scavenges, inset is degradation rate of ciprofloxacin. (Reproduced with permission from [80], copyright 2019, Elsevier) (**c**) Diagram of band gap structure of bulk and exfoliated g-C_3_N_4_ (**d**) Photocatalytic degradation of ciprofloxacin over g-C_3_N_4_ as well as exfoliated g-C_3_N_4_. (Reproduced with permission from [81], copyright 2019, Springer Nature).

**Figure 8 ijms-23-08130-f008:**
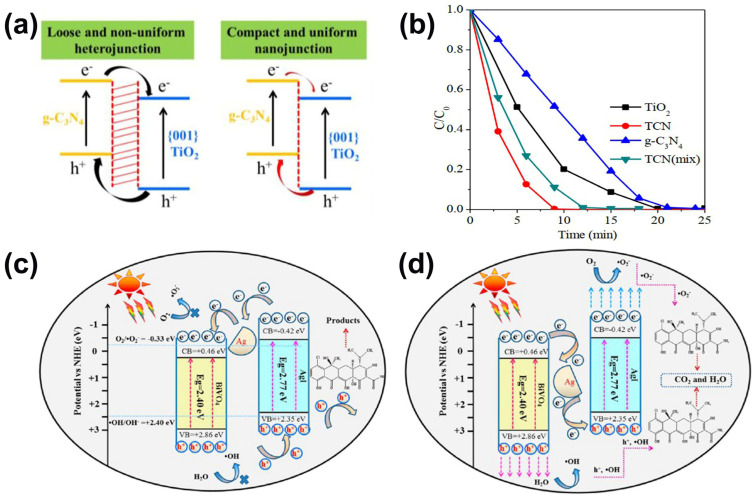
(**a**) Proposed heterojunction differences between TCN and TCN(mix) (**b**) Photocatalytic degradation efficiencies of tetracycline by employing TiO_2_, TCN, TCN (mix), and g-C_3_N_4_ as the photocatalysts under the xenon lamp irradiation. (Reproduced with permission from [87], copyright 2017, Elsevier). Schematic Illustration of the Mechanism for the Photocatalytic Degradation of tetracycline under Visible Light Irradiation over AgI/BiVO_4_ Nanocomposite: (**c**) Traditional Model and (**d**) Z-Scheme Heterojunction System. (Reproduced with permission from [88], copyright 2016, American Chemical Society).

**Figure 9 ijms-23-08130-f009:**
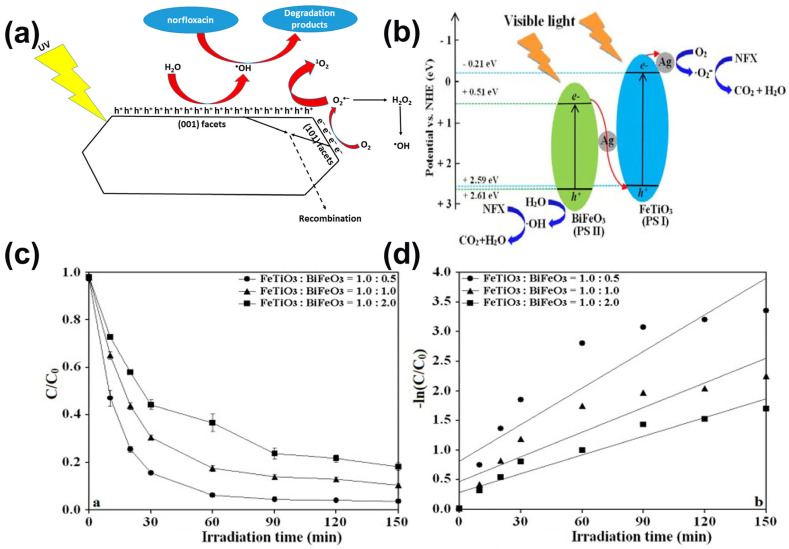
(**a**) Photocatalysis Mechanism of {001} Faceted TiO_2_/Ti Film. (Reproduced with permission from [92], copyright 2016, American Chemical Society) (**b**) Possible mechanism diagram on Z-scheme Ag/FeTiO_3_/Ag/BiFeO_3_ system (**c**) Effect of mass ratio of FeTiO_3_ and BiFeO_3_ and (**d**) degradation reaction kinetics on photocatalytic activity (2.0 wt.% Ag; 5.0 mg/L norfloxacin; 1.0 g/L catalyst). (Reproduced with permission from [93], copyright 2018, Elsevier).

**Figure 10 ijms-23-08130-f010:**
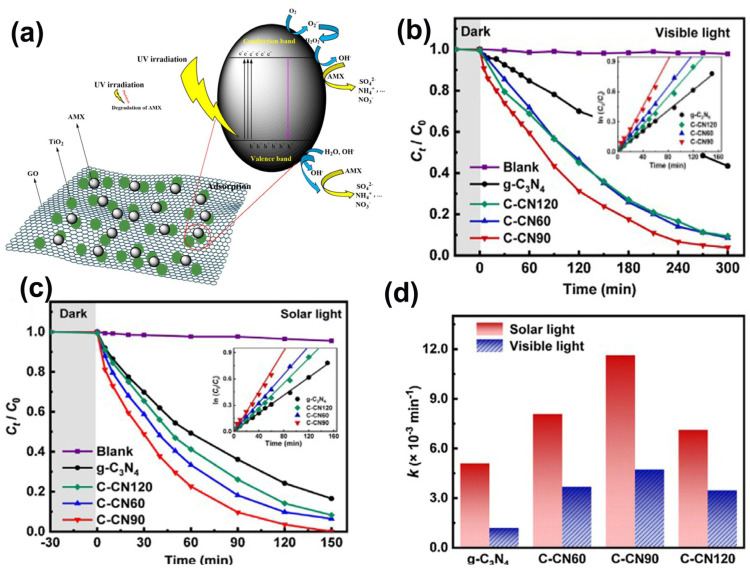
(**a**) Possible mechanism of amoxicillin degradation at GO/TiO_2_ surface. (Reproduced with permission from [97], copyright 2021, Springer Nature). Photocatalytic degradation kinetics of amoxicillin by the synthesized materials under (**b**) visible light and (**c**) simulate solar light. (**d**) amoxicillin degradation rate constants under solar and visible light. (Reproduced with permission from [99], copyright 2021, Elsevier).

**Table 1 ijms-23-08130-t001:** Comparison of the photocatalytic activity of different photocatalysts for antibiotic degradation.

Antibiotic	Catalyst	Results	Degradation Mechanism	Ref.
Ciprofloxacin	Zn-doped Cu_2_O	94.6% Ciprofloxacindegraded in 240 min	Mechanism by the •OH radical and h^+^	[80]
Ciprofloxacin	Exfoliated g-C_3_N_4_	78% Ciprofloxacindegraded in 60 min	Mechanism by the •O_2_^−^ radical and h^+^	[81]
Tetracycline	Heterostructured AgI/BiVO_4_	94.91% Tetracycline degraded in 60 min	Mechanism by the •OH, •O_2_^-^radical and h^+^	[88]
Tetracycline	Heterostructured TiO_2_/g-C_3_N_4_	100 mg TiO_2_/g-C_3_N_4_can decompose 2 mg Tetracycline in 9 min (2.2 mg/min)	Mechanism by the •O_2_^−^ radical and h^+^	[87]
Tetracycline	C–N–S-TiO_2_	>99% Tetracycline degraded in 360 min	Mechanism by the •O_2_^−^ radical and h^+^	[86]
Norfloxacin	Cu-doped BiOBr	96.5% Norfloxacin degraded in 150 min	Mechanism by the h^+^	[94]
Norfloxacin	Z-scheme Ag/FeTiO_3_/Ag/BiFeO_3_	Photocatalytic degradation rate of Norfloxacin is 0.64 × 10^−2^ min^−1^	Mechanism by the •OH radical and h^+^	[93]
Amoxicillin	Graphene Oxide/TiO_2_	91.25% Amoxicillin degraded in 60 min	Mechanism by the h^+^	[97]
Amoxicillin	Carbon-rich g-C_3_N_4_ nanosheets	Photocatalytic degradation rate of Amoxicillin is 0.47 × 10^−2^ min^−1^	Mechanism by the •O_2_^−^ radical	[99]

## Data Availability

No data.

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
