# Peer review of "Photocatalytic Degradation of Some Typical Antibiotics: Recent Advances and Future Outlooks"

_ijms, 2022, doi:10.3390/ijms23158130_

Round 1

Reviewer 1 Report

The manuscript examined the Photocatalytic Degradation of Some Typical Antibiotics: Recent advances and Future Outlooks. The topic addressed in the manuscript falls within the scope of International Journal of Molecular Sciences. Generally, the study was well conceived and presented. Nonetheless, few corrections need to be made. Thus, I recommend the acceptance of the manuscript for publication subject to a minor revision. 

Throughout the manuscript, references at the end of a sentence should be cited before full stop. 

Line 205: Delete ‘be’

Line 241: Replace ‘was’ with ‘is’

Line 282: Write ‘Yu et al’ as ‘Yul et al [72] and delete [72] at the end of the sentence. 

Line 289: Write ‘Malakootian et al’ as ‘Malakootian et al [69]’ and delete [69] at the end of the sentence. 

Line 294: ‘Pattnaik et al’ as ‘Pattnaik et al [73]’ and delete [73] at the end of the sentence. 

Line 356: ‘Sayed et al’ as ‘Sayed et al [84]’ and delete [84] at the end of the sentence. 

Line 361: ‘Tang et al’ as ‘Tang et al [85]’ and delete [85] at the end of the sentence. 

Line 367: ‘Lv et al’ as ‘Lv et al [86]’ and delete [86] at the end of the sentence. 

Line 375: ‘Tang et al’ as ‘Tang et al [87]’ and delete [87] at the end of the sentence. 

Line 402: ‘Mirzaei et al’ as ‘Mirzaei et al [91]’ and delete [91] at the end of the sentence. 

Line 408: ‘Huang et al’ as ‘Huang e

t al [92]’

Author Response

Response to the reviewers’ comments

We would like to express our appreciation for the time devoted to our manuscript and for the helpful comments we received which accordingly enabled us to improve the manuscript. Please find below a point-by-point response to the comments received from the reviewer. The main changes to the manuscripts are described here. Further minor changes are highlighted in track-mode in the revised manuscript.

Thank you very much for your consideration.

Best regards,

Yuheng Wang, on behalf of the co-authors.

Reviewer #1

The manuscript examined the Photocatalytic Degradation of Some Typical Antibiotics: Recent advances and Future Outlooks. The topic addressed in the manuscript falls within the scope of International Journal of Molecular Sciences. Generally, the study was well conceived and presented. Nonetheless, few corrections need to be made. Thus, I recommend the acceptance of the manuscript for publication subject to a minor revision. Throughout the manuscript, references at the end of a sentence should be cited before full stop. 

Response: We thank the reviewer for her or his suggestions of the manuscript

(1) Line 205: Delete ‘be’

Line 241: Replace ‘was’ with ‘is’

Line 282: Write ‘Yu et al’ as ‘Yul et al [72] and delete [72] at the end of the sentence. 

Line 289: Write ‘Malakootian et al’ as ‘Malakootian et al [69]’ and delete [69] at the end of the sentence. 

Line 294: ‘Pattnaik et al’ as ‘Pattnaik et al [73]’ and delete [73] at the end of the sentence. 

Line 356: ‘Sayed et al’ as ‘Sayed et al [84]’ and delete [84] at the end of the sentence. 

Line 361: ‘Tang et al’ as ‘Tang et al [85]’ and delete [85] at the end of the sentence. 

Line 367: ‘Lv et al’ as ‘Lv et al [86]’ and delete [86] at the end of the sentence. 

Line 375: ‘Tang et al’ as ‘Tang et al [87]’ and delete [87] at the end of the sentence. 

Line 402: ‘Mirzaei et al’ as ‘Mirzaei et al [91]’ and delete [91] at the end of the sentence. 

Line 408: ‘Huang et al’ as ‘Huang e

t al [92]’.

Response: The revision related to your comments of minor revisions were all updated in the whole paper.

Reviewer 2 Report

The authors have submitted an important review article entitled "Photocatalytic Degradation of Some Typical Antibiotics: Recent advances and Future Outlooks" which summarized strategies to photoremediate antibiotic-contaminated wastewaters. The manuscript reads well overall; But, it will need a spelling and style check. Although some similar review papers have been published in this area, this manuscript can be considered for publication after a major revision.

Comments:

1- First of all, I would like to recommend authors design an informative and simple “Graphical Abstract” for this study to better show the whole story at a glance.

2- Please carefully revise the manuscript to remove the grammatical errors and vague sentences and try to avoid unnecessary information (For example: To degrade these residuals, a variety of photocatalysts can be utilized to address this issue …. To degrade these residuals, a variety of photocatalysts can be utilized.). I am not quite sure why the authors have cited references at the beginning of the next sentence! (“Antibiotics are chemotherapeutic agents that retard and eradicate bacterial infections. [1]” should be “Antibiotics are chemotherapeutic agents that retard and eradicate bacterial 33 infections [1]. “)

Please double-check the whole manuscript.

3- The novelty statement of the article poorly represents the work and needed to be developed to highlight the importance of this work and how it is different from recently published reports with similar titles such as “A review on photocatalysis in antibiotic wastewater: Pollutant degradation and hydrogen production, Chinese Journal of Catalysis Volume 41, Issue 10, October 2020, Pages 1440-1450” or “Recent advances in photodegradation of antibiotic residues in water, Chemical Engineering Journal

Volume 405, 1 February 2021, 126806”.

4- Metal-organic frameworks (MOFs) are a big family of nanoparticles with intriguing characteristics which have been exploited as robust photoactive agents to eliminate antibiotics from aqueous media. Recently, several interesting review papers have been also published in this regard, Please read some impactful ones and add them to your manuscript (For example A review of metal-organic framework (MOFs)-based materials for antibiotics removal via adsorption and photocatalysis, Chemosphere Volume 272, June 2021, 129501).

6- It is expected that at least one master table is provided by the authors of this work to compare the photo-degradability of different materials which has been reviewed in this manuscript. Please tabulate a comparative table containing recently published papers.

7- To better explain and support part 3 of this review paper: “Principle and fundamental mechanism of photocatalytic degradation of antibiotics” Please read the following publications and add invaluable information to your manuscript (10.1016/B978-0-12-818806-4.00013-9 and 10.1016/B978-0-12-818806-4.00010-3).

8- In the last section, first of all, I suggest not citing references and providing independent conclusions. Furthermore, based on the tile, please revise this section to something like “Conclusions and Future perspective”. The last paragraph of the conclusion is rarely considered as a future perspective. So, please provide more insights related to future trends of photodegradation-based treatment of antibiotic-polluted water. 

Author Response

Response to the reviewers’ comments

We would like to express our appreciation for the time devoted to our manuscript and for the helpful comments we received which accordingly enabled us to improve the manuscript. Please find below a point-by-point response to the comments received from the reviewer. The main changes to the manuscripts are described here. Further minor changes are highlighted in track-mode in the revised manuscript.

Thank you very much for your consideration.

Best regards,

Yuheng Wang, on behalf of the co-authors.

Reviewers' Comments to Author:

Reviewer #2

The authors have submitted an important review article entitled "Photocatalytic Degradation of Some Typical Antibiotics: Recent advances and Future Outlooks" which summarized strategies to photoremediate antibiotic-contaminated wastewaters. The manuscript reads well overall; But, it will need a spelling and style check. Although some similar review papers have been published in this area, this manuscript can be considered for publication after a major revision.

Response: We thank the reviewer for her or his suggestions of the manuscript.

(1) First of all, I would like to recommend authors design an informative and simple “Graphical Abstract” for this study to better show the whole story at a glance.   

Response: We agree with this comment and added aGraphical Abstract” figure in Abstract on page 2.

Changes: The suggested Graphical Abstract” figure was added (pages 3) in the abstract (check in the revised paper):

(2) Please carefully revise the manuscript to remove the grammatical errors and vague sentences and try to avoid unnecessary information (For example: To degrade these residuals, a variety of photocatalysts can be utilized to address this issue …. To degrade these residuals, a variety of photocatalysts can be utilized.). I am not quite sure why the authors have cited references at the beginning of the next sentence! (“Antibiotics are chemotherapeutic agents that retard and eradicate bacterial infections. [1]” should be “Antibiotics are chemotherapeutic agents that retard and eradicate bacterial 33 infections [1]. “)

Response: We indeed made mistakes on grammatical errors and vague sentences, particularly on putting references at wrong locations. These mentioned mistakes in whole text has been amended accordingly.

(3) The novelty statement of the article poorly represents the work and needed to be developed to highlight the importance of this work and how it is different from recently published reports with similar titles such as “A review on photocatalysis in antibiotic wastewater: Pollutant degradation and hydrogen production, Chinese Journal of Catalysis Volume 41, Issue 10, October 2020, Pages 1440-1450” or “Recent advances in photodegradation of antibiotic residues in water, Chemical Engineering Journal Volume 405, 1 February 2021, 126806”.

Response: The novelty statement related to difference with other recent published reports were updated to highlight the importance of this work.

Changes: The suggested additional discussion was added (pages 3) in the main manuscript as follow:

Pages 3:

‘The photodegradation of antibiotic pollutants were already reviewed recently [5,15]. However, the knowledge of the critical degradation mechanisms and underlying reaction pathways of some typical catalysts in photodegradation reaction of antibiotics requires deeper discussion. Furthermore, a comprehensive overview on the possible sources and dangers of the antibiotics pollutants released through the ecological chain, in particular including the consequences and damages of antibiotics residuals on the environment and human health, is still missing. In addition, the overall introduction of some commonly-used photocatalytic nanomaterials and their application in degradation of some typical antibiotics is essential to confirm their practical superiority and effectiveness as photodegradation catalysts.’

(4) Metal-organic frameworks (MOFs) are a big family of nanoparticles with intriguing characteristics which have been exploited as robust photoactive agents to eliminate antibiotics from aqueous media. Recently, several interesting review papers have been also published in this regard, Please read some impactful ones and add them to your manuscript (For example A review of metal-organic framework (MOFs)-based materials for antibiotics removal via adsorption and photocatalysis, Chemosphere Volume 272, June 2021, 129501).

Response: The addition related to MOFs-based photocatalysts were updated in this work.

Changes: The suggested additional discussion was added (pages 8) in the main manuscript as follow:

Page 8:

‘4.4. Metal-organic frameworks (MOFs)-based photocatalysts

Metal-organic frameworks (MOFs), as a new class of coordination polymers with periodic network structures formed by the self-assembly between metal ions/metal clusters and organic ligands [66]. By modifying linkers employing functional groups, highly porous structures with a remarkable surface area could be obtained with tuned surface structures [67]. MOFs were first discovered in the mid-1990’s by Omar Yaghi and the invention of novel MOFs promised long-lived influence in the area of chemistry, physics, biology and material sciences, in particular extensively used in photocatalysis due to their high surface area, adjustable porosity and pore volume [6769]. Thus, MOFs are promising to be highly-effective materials for photocatalytic degradation of antibiotics in solution [66,70]. Although various MOFs-based materials have been utilized to remove antibiotics, the development of more efficient degradation agents remains a key problem facing more active MOFs-based photocatalytic degradation with more active sites and large surface area with group functionalization [66,67]. According to the previous report, the organic linker serves as the VB while the metallic cluster act as CB. Under exposure to light, MOFs behave like semiconductors, thus they can be deemed as a potential photocatalyst for highly effective degradation of antibiotics due to superior high thermal and mechanical stability and excellent structural characteristics [66].’

(5) It is expected that at least one master table is provided by the authors of this work to compare the photo-degradability of different materials which has been reviewed in this manuscript. Please tabulate a comparative table containing recently published papers

Response: The additional table related to comparison among different photocatalysts were updated in this work.

Changes: The suggested additional table was added (pages 16) in the main manuscript as follow:

Page 16:

Antibiotic

Catalyst

Results

Degradation Mechanism

Ref.

Ciprofloxacin

Zn-doped Cu2O

94.6% Ciprofloxacin degraded in 240 min

Mechanism by the •OH radical and h+

[80]

Ciprofloxacin

Exfoliated g-C3N4

78% Ciprofloxacin degraded in 60 min

Mechanism by the •O2- radical and h+

[81]

Tetracycline

Heterostructured AgI/BiVO4

94.91% Tetracycline degraded in 60 min

Mechanism by the •OH, •O2-radical and h+

[88]

Tetracycline

Heterostructured TiO2/g-C3N4

100 mg TiO2/g-C3N4 can decompose 2 mg Tetracycline in 9 min (2.2 mg/min)

Mechanism by the •O2- radical and h+

[87]

Tetracycline

C–N–S-TiO2

99% Tetracycline degraded in 360 min

Mechanism by the •O2- radical and h+

[86]

Norfloxacin

Cu-doped BiOBr

96.5% Norfloxacin degraded in 150 min

Mechanism by the h+

[94]

Norfloxacin

Z-scheme Ag/FeTiO3/Ag/BiFeO3

Photocatalytic degradation rate of Norfloxacin is 0.64 ×10-2 min-1

Mechanism by the •OH radical and h+

[93]

Amoxicillin

Graphene Oxide/TiO2

91.25% Amoxicillin degraded in 60 min

Mechanism by the h+

[98]

Amoxicillin

Carbon-rich g-C3N4 nanosheets

Photocatalytic degradation rate of Amoxicillin is 0.47 ×10-2 min-1

Mechanism by the •O2- radical

[100]

(6) To better explain and support part 3 of this review paper: “Principle and fundamental mechanism of photocatalytic degradation of antibiotics” Please read the following publications and add invaluable information to your manuscript (10.1016/B978-0-12-818806-4.00013-9 and 10.1016/B978-0-12-818806-4.00010-3).

Response: The addition related to reaction mechanism were updated in this work

Changes: The suggested addition was added (pages 5 and 6) in the main manuscript as follow:

Page 5 and 6:

‘Considering the prediction of application and efficiency of a type of photocatalytic material, optical bandgap (Eg) is a very important factor to evaluate the ability of photoabsorption and photocatalytic efficiency. Mehrorang et al. put forward the method and criterion on measurement of bandgap and divided the concept of bandgap into two categories of photonic and electrochemical bandgap facing polyfluorene co-polymers as photocatalysts [34,35]. In addition, They concluded that as a result that preventing charge recombination which accordingly leads to a higher lifetime of the active holes would upgrade their activity of degradation on antibiotics, a great strategy to enhance the activity of photocatalysts under visible light is related to interfacial charge transfer from a separate energy surface to molecular continuous ones from solids [34,35].’

(7) In the last section, first of all, I suggest not citing references and providing independent conclusions. Furthermore, based on the tile, please revise this section to something like “Conclusions and Future perspective”. The last paragraph of the conclusion is rarely considered as a future perspective. So, please provide more insights related to future trends of photodegradation-based treatment of antibiotic-polluted water.

Response: The addition about future insights related to future trends of photodegradation-based treatment of antibiotic-polluted water were updated in this work and we removed references in this section.

Changes: The suggested addition and revision was added (pages 16 and 17) in the main manuscript as follow:

Page 16 and 17:

‘In this review, the photocatalytic degradation of antibiotics was summarized. Firstly, the mechanism of photocatalytic degradation of antibiotics depending on the formation of free radicals and reactive oxygen species and the consequences of antibiotics in wastewater on the environment and human health were reviewed. Some photocatalytically widely used antibiotics were then concluded and as well as the commonly used photocatalysts were also introduced. Heteroatom-doping is generally used as a strategy to enhance the photocatalytic performance of a photocatalyst, particularly metal atoms as dopants. However, it should be noted that metal dopants could serve as recombination centers at higher concentrations which reduce the efficiency of a photocatalyst. In case of that, future research should also focus on other options including doping with non-metals such as nitrogen, boron, sulfur and phosphorus. Meanwhile, the formation of heterojunction with other semiconductors can also play a significant role in the modification of photocatalysts on the degradation of antibiotics due to other semiconductors can possibly serve as photosensitizers while simultaneously inhibiting electron-hole recombination. Thus, these methods can afford visible lightdriven photocatalysts with enhanced photocatalytic activity by either narrowing the band gap of the photocatalyst or improving charge separation.

Physicochemical properties such as morphology and surface areas are also very critical factors in the performance of catalysts during photodegradation studies. As mentioned above, further exploring photocatalysts with different morphologies and surface areas can effectively enhance the performance of catalysts. Furthermore, the degradation pathway also provides a clear introduction to the fate and transformation of antibiotics during the photocatalytic degradation process. Thus, exploring the photocatalytic degradation mechanism in-depth at the atomic level is also compulsory for promoting the efficiency of antibiotic degradation.

The use of solar radiation and visible light sources to activate the photocatalysts during photodegradation of antibiotics such as ciprofloxacin, tetracycline, norfloxacin and amoxicillin is still limited. Therefore, the exploration and development of photodegradation induced by UV light sources are still in urgent demand.

Firstly,in the long term, although the removal rate of antibiotics is still being optimized, the removal rate of the chemical oxygen demand is yet relatively high during the degradation of antibiotics by photocatalysts and thus it confirms that the mineralization degree of antibiotics is needed to be optimal. There are many intermediates during the process of photodegradation and a deep study into intermediates is therefore also critical to improving the performance of catalysts. Secondly, in experiments, it involves regular and constant stirring to prevent the agglomeration of materials in media during the degradation of antibiotics by photocatalysts, requiring additional energy consumption. Thirdly, the problem of antibiotics is not only the problems are induced by water quality but also the accumulation of antibiotics in water will lead to the generation of microbial resistance genes. There is currently still a lack of research on photocatalysts’ limited resistance genes. Finally, The recycling ability of a photocatalyst is a significant index to evaluate its cost-effectiveness and feasibility for practical applications in the degradation of antibiotics. To minimize any possible waste, it is preferred to design photocatalysts that have quasi-same photoactivity during each cycle. Also, it is important to design photocatalysts that are easier to separate and recycle in order not to lose any worthy materials during the photocatalytic reaction and thus the separation of photocatalysts from the aqueous phase is quite crucial under an economic standpoint. It is noted that the operating cost of the photocatalytic reaction mainly originates from the single-use photocatalyst without recycling. Regarding the repetitive usage of photocatalysts, deep research on how to sustainably use photocatalysts with many cyclings for degradation of antibiotics are still in urgent demand.’

Reviewer 3 Report

In this paper, the authors did a short revision about photocatalytic degradation of some antibiotics. The papers is well written, however the quality of the pictures is very bad and should be improved. In addition, the size of the structure in figure 6 should be reduced. On the reference section, the numeration is doubled and this should be corrected.

Some important reference from high impact journal as for example 10.1016/j.apcatb.2020.119556 are missing and should be added.

Author Response

Response to the reviewers’ comments

We would like to express our appreciation for the time devoted to our manuscript and for the helpful comments we received which accordingly enabled us to improve the manuscript. Please find below a point-by-point response to the comments received from the reviewer. The main changes to the manuscripts are described here. Further minor changes are highlighted in track-mode in the revised manuscript.

Thank you very much for your consideration.

Best regards,

Yuheng Wang, on behalf of the co-authors.

Reviewer #3

In this paper, the authors did a short revision about photocatalytic degradation of some antibiotics. The papers is well written, however the quality of the pictures is very bad and should be improved. In addition, the size of the structure in figure 6 should be reduced. On the reference section, the numeration is doubled and this should be corrected. Some important reference from high impact journal as for example 10.1016/j.apcatb.2020.119556 are missing and should be added.

Response: We thank the reviewer for her or his suggestions of the manuscript. The revision related to your comments of revisions on each figures were all updated in the whole paper. Meanwhile, I am so sorry that the reference you mentioned is not suitable for this paper, because my review just focuses on four typical antibiotics which were not involved in the paper you pointed out. I hope you can understand this. I am very thankful for your understanding.

Changes: The suggested revision of figure 6 was added (pages 10) and the addition of reference in the manuscript (check in the paper).

Reviewer 4 Report

The authors have submitted an interesting article with the title “Photocatalytic Degradation of Some Typical Antibiotics: Recent advances and Future Outlooks”. I recommend the publication of this work after a major revision.

1)      There are several methods to remove the antibiotic from the environment(Especially aqueous media). In the abstract part, the author should explain more why they select photodegradation over other approaches.

2)      The recycling ability of a photocatalyst is an important parameter to evaluate its cost-effectiveness and feasibility for real-world applications. Usually, photocatalysts show high stability after 3 to 5 times recycling.

3)      The by-products of pharmaceutical degradation may be more harmful to the ecosystem. Add date about degradation pathway of mentioned drugs if there is any available information.

Author Response

Response to the reviewers’ comments

We would like to express our appreciation for the time devoted to our manuscript and for the helpful comments we received which accordingly enabled us to improve the manuscript. Please find below a point-by-point response to the comments received from the reviewer. The main changes to the manuscripts are described here. Further minor changes are highlighted in track-mode in the revised manuscript.

Thank you very much for your consideration.

Best regards,

Yuheng Wang, on behalf of the co-authors.

Reviewer #4

The authors have submitted an interesting article with the title “Photocatalytic Degradation of Some Typical Antibiotics: Recent advances and Future Outlooks”. I recommend the publication of this work after a major revision.

Response: We thank the reviewer for her or his suggestions of the manuscript.

(1) There are several methods to remove the antibiotic from the environment(Especially aqueous media). In the abstract part, the author should explain more why they select photodegradation over other approaches.

Response: The addition related to the reason about why researchers select photodegradation over other approaches was updated in abstract.

Changes: The suggested addition and revision was added (pages 1) in the main manuscript as follow:

Page 1:

‘Currently, there are some most popular techniques for removing antibiotics pollutants from water including physical adsorption, flocculation and chemical oxidation, however, they usually leave plenty of chemical reagents and polymer electrolytes in water leading to difficulty in post-treatment of unmanageable deposits. Furthermore, regarding some advantages on cost-effectiveness, remarkable efficiency, facile conditions for reactions, and nontoxicity of photocatalysts to degradation of antibiotics, a variety of photocatalysts can be utilized to degrade these residuals to address this issue. Thus urgent need for effectively and rapidly photocatalytic degradation led to an increased interest in finding more sustainable catalysts for antibiotic degradation.’

(2) The recycling ability of a photocatalyst is an important parameter to evaluate its cost-effectiveness and feasibility for real-world applications. Usually, photocatalysts show high stability after 3 to 5 times recycling.

Response: The addition related to discussion about recycling ability was updated in conclusion section.

Changes: The suggested addition and revision was added (pages 17) in the main manuscript as follow:

Page 17:

‘Finally, The recycling ability of a photocatalyst is a significant index to evaluate its cost-effectiveness and feasibility for practical applications in the degradation of antibiotics. To minimize any possible waste, it is preferred to design photocatalysts that have quasi-same photoactivity during each cycle. Also, it is important to design photocatalysts that are easier to separate and recycle in order not to lose any worthy materials during the photocatalytic reaction and thus the separation of photocatalysts from the aqueous phase is quite crucial under an economic standpoint. It is noted that the operating cost of the photocatalytic reaction mainly originates from the single-use photocatalyst without recycling. Regarding the repetitive usage of photocatalysts, deep research on how to sustainably use photocatalysts with many cyclings for degradation of antibiotics are still in urgent demand.’

(3) The by-products of pharmaceutical degradation may be more harmful to the ecosystem. Add date about degradation pathway of mentioned drugs if there is any available information.

Response: Dear reviewer, as for by-products, yes, they are of course very importantto deeply studied when they maybe do more harm to the ecosystem. However, among the examples I discussed in this review, their work basically avoided to mention by-products produced during photocatalytic degradation of antibiotics.So I can not provide you the details about by-products of antibiotics degradation from the cases in this review. I noticed that you mentioned degradation pathway of mentioned drugs in report and I have added a interesting table to compare with each other among the different photocatalysts including comparison on degradation pathway. Please check the table below.

Changes: The table related to comparison on degradation pathway of mentioned antibiotics was added (pages 16) in the main manuscript as follow:  

Page 16:

Antibiotic

Catalyst

Results

Degradation Mechanism

Ref.

Ciprofloxacin

Zn-doped Cu2O

94.6% Ciprofloxacin degraded in 240 min

Mechanism by the •OH radical and h+

[80]

Ciprofloxacin

Exfoliated g-C3N4

78% Ciprofloxacin degraded in 60 min

Mechanism by the •O2- radical and h+

[81]

Tetracycline

Heterostructured AgI/BiVO4

94.91% Tetracycline degraded in 60 min

Mechanism by the •OH, •O2-radical and h+

[88]

Tetracycline

Heterostructured TiO2/g-C3N4

100 mg TiO2/g-C3N4 can decompose 2 mg Tetracycline in 9 min (2.2 mg/min)

Mechanism by the •O2- radical and h+

[87]

Tetracycline

C–N–S-TiO2

99% Tetracycline degraded in 360 min

Mechanism by the •O2- radical and h+

[86]

Norfloxacin

Cu-doped BiOBr

96.5% Norfloxacin degraded in 150 min

Mechanism by the h+

[94]

Norfloxacin

Z-scheme Ag/FeTiO3/Ag/BiFeO3

Photocatalytic degradation rate of Norfloxacin is 0.64 ×10-2 min-1

Mechanism by the •OH radical and h+

[93]

Amoxicillin

Graphene Oxide/TiO2

91.25% Amoxicillin degraded in 60 min

Mechanism by the h+

[98]

Amoxicillin

Carbon-rich g-C3N4 nanosheets

Photocatalytic degradation rate of Amoxicillin is 0.47 ×10-2 min-1

Mechanism by the •O2- radical

[100]

Round 2

Reviewer 2 Report

The manuscript is well amended and it is ready to be published. I have no further comments.

Reviewer 4 Report

The authors replied to my comments and I have no further comments.